# Beyond the Extracellular Vesicles: Technical Hurdles, Achieved Goals and Current Challenges When Working on Adipose Cells

**DOI:** 10.3390/ijms22073362

**Published:** 2021-03-25

**Authors:** María Gómez-Serrano, Viviane Ponath, Christian Preußer, Elke Pogge von Strandmann

**Affiliations:** Institute for Tumor Immunology, Center for Tumor Biology and Immunology (ZTI), Philipps University of Marburg, 35043 Marburg, Germany; ponath@staff.uni-marburg.de (V.P.); preusserc@staff.uni-marburg.de (C.P.); poggevon@staff.uni-marburg.de (E.P.v.S.)

**Keywords:** adipocytes, differential centrifugation, extracellular vesicles, isolation methods, primary culture, single-vesicle analysis

## Abstract

Adipose tissue and its crosstalk with other organs plays an essential role in the metabolic homeostasis of the entire body. Alteration of this communication (i.e., due to obesity) is related to the development of several comorbidities including type 2 diabetes, cardiovascular diseases, or cancer. Within the adipose depot, adipocytes are the main cell type and thus the main source of secreted molecules, which exert modulating effects not only at a local but also at a systemic level. Extracellular vesicles (EVs) have recently emerged as important mediators in cell–cell communication and account for part of the cellular secretome. In recent years, there has been a growing body of research on adipocyte-derived extracellular vesicles (Ad-EVs). However, there is still a lack of standardized methodological approaches, especially regarding primary adipocytes. In this review, we will provide an outline of crucial aspects when working on adipose-derived material, with a special focus on primary adipocytes. In parallel, we will point out current methodological challenges in the EV field and how they impact the transcriptomic, proteomic and functional evaluations of Ad-EVs.

## 1. Introduction

Adipocytes and extracellular vesicles (EVs) have more in common than expected. Although they differ by approximately 1000 times in terms of size (adipocytes have average diameters of 100–120 μm, whereas EVs range from 40 to 200 nm and up to 1000 nm), both are lipid-based biological entities, have shown to play an important modulatory role in other cells and have experienced a similar history regarding the growing interest in the scientific community. 

For a long time, adipose tissue was considered only as a fat reservoir. In fact, in the 1940s it was regarded as a type of connective tissue where fat could be accumulated in the form of lipid droplets and it was not until the 1980s when its key role in metabolic homeostasis was recognized [1]. Over that decade, several molecules secreted by adipocytes were determined and finally, in 1994, thanks to the discovery of leptin, adipose tissue was recognized as a true endocrine organ [2]. Although adipose tissue is composed of other cell types such as fibroblasts or macrophages, adipocytes are its main cell type [3,4] and their distinctive characteristics contribute to the three-dimensional (3D) structure of this tissue [5]. Moreover, the list of hormones secreted by adipocytes has been increasing over the years [6], which highlights the pleiotropic regulatory role of this tissue [7,8]. 

EVs were first referred to as “platelet dust” by Peter Wolf in 1967 [9] and were regarded as useless cellular trash resulting from the blood coagulation process. Some decades later, pioneering studies from Harding et al. [10] and Pan et al. [11,12] recognized a new form of EVs (called exosomes) arising from budding of the intracellular endosomal membranes. 

The general term “EVs” comprises nanoparticles released by virtually all cell types, surrounded by a lipid bilayer and with distinct cargoes of proteins, nucleic acids, and lipids, which can modulate a recipient cell. Currently, EVs are classified into different subclasses based on their different biophysical properties on the one hand and their biogenesis routes on the other hand. Per se, all secreted membrane-enclosed particles are part of the umbrella term of EVs, including those secreted during programmed cell death (apoptotic bodies, ranging from 50 to 5 μm). The scientific community mostly focuses on the two main classes of EVs—namely, microvesicles or ectosomes (100–1000 nm), which directly bud from the plasma membrane, and the already mentioned exosomes (40–200 nm), which are of endosomal origin (for review see [13]). Their distinction is largely based on the different biogenesis routes; however, both types of vesicles share their biophysical and biochemical properties to some extent, such as size and surface marker expression [14,15]. Nevertheless, several surface membrane-bound markers such as the tetraspanin CD63 as well as other intravesicular biogenesis factors such as ALIX or syntenin-1, allow discrimination between the two subclasses (see section on Minimal information for studies of extracellular vesicles (MISEV) guidelines). In total, a plethora of names for different EVs can be found in the literature, so that the term “extracellular vesicle” or “EV” has now been agreed on by the international community as the most suitable term [16].

Publications related to “adipose tissue” and “extracellular vesicles” show similar trends, reflecting the growing research interest as revealed by published articles on the PubMed database (Figure 1). Nowadays, both topics show an ever-increasing interest among the research community and overlaps between both areas have been discovered. Interestingly, during the last decade, EVs released by adipocytes have been recognized as essential parts of their secretome [17], exerting important modulatory roles in a broad spectrum of cells (reviewed in [18,19]). As a result, the combination of both topics has followed a similar trend during the last decade, currently being covered by more than 200 publications (Figure 1). 

Despite the growing body of research working on adipose-derived material, most of the approaches have relied on in vitro models of differentiated adipocytes (extensively reviewed in [22,23]) and very few references have been focused on adipocyte-derived EVs (Ad-EVs) from primary ex vivo sources [24,25,26,27,28,29,30]. Moreover, methodological standardization in the field is still lacking and only a few recent articles have displayed fully described approaches [25,27,28,29,30,31]. To our knowledge, commentaries addressing this question have not been published so far. Therefore, in this review, we will provide an overview of the essential considerations when working with adipose tissue based on the existing literature on Ad-EVs. We will mainly focus on the methodological aspects described for primary adipocytes, highlighting their strengths and weaknesses and their potential future implementations in transcriptomic, proteomic, and functional approaches. 

## 2. The Importance of Standardization: MISEV Guidelines

Several studies have highlighted EVs as important carriers of signaling molecules representing a promising new tool in clinical research as potential biomarkers or drug carriers. These new research areas are reflected in the ever-increasing number of EV-related publications over the last two decades and necessitate standardization of methods concerning EV isolation/separation, characterization, and use in functional studies. In 2014, the “Minimal information for studies of extracellular vesicles” (MISEV) guidelines were published and were further updated in 2018 [16]. Here, experts of the International Society for Extracellular Vesicles (ISEV) outlined the essential requirements for transparent and reliable EV research. The guidelines recommend describing EVs by different parameters such as (a) physical characteristics such as size (small EVs <100–200 nm, medium/large >200 nm) or density; (b) biochemical composition (protein expression on their surface—e.g., CD9, CD63, CD81, annexin V, etc.); and (c) the origin of their release, including information about the culture conditions or the producer cell, among others [16]. 

There are several methods available to quantify the size and concentration of EVs with a rapidly evolving variety of single-EV characterization high-resolution and throughput approaches [32]. Preanalytical parameters such as the purification method and the EV-producing entity need to be considered when analyzing the size. It should be noted that purification methods such as precipitation or differential ultracentrifugation (dUC) can lead to the isolation of differently sized EV populations and co-isolation of non-EV structures [16]. Thus, the guidelines recommend using two different techniques for assessing single EVs. Historically, the most common but technically demanding method was electron microscopy (especially transmission electron microscopy—TEM) and related techniques such as scanning probe microscopy (e.g., atomic force microscopy—AFM). Other methods using light-scattering pattern techniques such as nanoparticle tracking analysis (NTA), high-resolution flow cytometry (e.g., nanoflow cytometry—nFCM), multiangle light scattering coupled with asymmetric flow field-flow fractionation (AF4), or fluorescence-correlated spectroscopy (FSC) are also highly advisable. However, every method has drawbacks in certain aspects such as throughput, overestimation or underestimation of size, limited concentration or size detection range (reviewed in [32,33,34]), among others. To name one example: light-scattering techniques such as NTA regularly overestimate EV counts as it is not specific to EVs but also detects co-isolated particles such as protein aggregates [35,36]. Other co-isolated artefacts that need to be considered in adipocyte preparations are lipid droplets from broken cells. An in-depth description of isolation methods for EVs is beyond the scope of this review, and therefore we refer the readers to another article within this Special Issue [34]. In the case of EVs derived from adipose cells or tissues, standardized and well-documented protocols (as far as they are available) should be surveyed, ideally following the MISEV guidelines. These procedures should be adapted based on follow-up studies to achieve a reasonable compromise between quality and quantity according to the goal of the study. In particular, the advent of novel technologies such AF4 or those newly adapted for EV isolation such as free-flow electrophoresis (FFE) will help to further facilitate the purification of EVs from different source materials [37,38], including adipocytes.

Quantification of the EV concentration constitutes another common challenge. Total protein amount is frequently used to quantify EVs and can be measured by colorimetric assays such as the bicinchoninic acid assay (BCA). One drawback here is the potential contamination of the EV sample with coprecipitated protein aggregates depending on the isolation method (e.g., dUC), resulting in an overestimation of protein content and, thus, EVs. Quantification of EV cargo such as total RNA [39] or total lipids [40,41,42] have also been described but remain technically difficult and often require special equipment. Lipid-based quantification may be particularly challenging when working with adipose-derived material due to the risk of artefacts (i.e., lipid droplets). Other methods of quantification focus on the detection of specific molecules on the EV surface—e.g., via ELISA or bead-based flow cytometry [33]. These assays usually detect tetraspanins such as CD63 or CD81, therefore limiting the quantification to certain types of EVs. 

To characterize the nature and purity of isolated EVs, it is recommended that EV-specific markers are detected [16]. On the one hand, this should include at least one transmembrane or GPI-anchored protein of non-tissue specific origin that is common in lipid bilayer structures such as EVs (e.g., the already cited tetraspanins CD9, CD63, CD81, and CD82, MHC class I or integrins). In addition, when analyzing tissue-specific EVs, cell type or tissue-specific markers need to be included—e.g., ERBB2 for breast cancer or EPCAM for epithelium. For Ad-EVs, perilipin-1 (PLIN1) has recently been defined as an evident marker [23]; however, some authors have already raised some concerns as PLIN1 localizes in lipid droplets (a potential co-isolated artefact) [43]. On the other hand, cytosolic proteins with lipid or membrane-binding ability that demonstrate the enclosure of intracellular material are also a requirement. These include proteins of the EV biogenesis pathway (i.e., the ESCRT complex and its associated factors Tsg101, Alix, flotillin-1, and Hsp70). In the context of Ad-EV research, the detection of fatty acid-binding protein 4 (FABP4) has been established as a cytoplasmic marker of choice (reviewed in [23]) despite its non-relationship with the ESCRT complex or associated factors.

Depending on the source material, EV composition can be heterogeneous and contain impurities that need to be ascertained. In tissue samples, for example, EVs can be released from different cell types, and mechanical destruction during the harvest of the material or cell death during cultivation can lead to contamination of the purified EV preparations. In biofluids (e.g., serum, milk, or urine), the origin of EVs is unknown and co-isolation of non-EV lipid particles is a common problem [16]. To exclude or at least characterize the cross-contamination with co-isolated structures such as lipoproteins (e.g., LDL or HDL), or with other cell organelles (e.g., nucleus, mitochondria, endoplasmic reticulum), negative markers need to be included. Some examples are but not restricted to histones and lamins, cytochrome c, and calnexin for the secretory pathway of the ER. If EVs are to be used for functional assays, co-isolated luminal and secreted proteins such as cytokines (e.g., TGFβ, interleukins), growth factors (e.g., VEGF), and extracellular matrix components (e.g., collagen, galectin-3-binding protein) need to be excluded as they can associate with EVs during the isolation procedure and influence the results of functional studies in an EV-independent manner.

Despite these recommendations, the isolation of EVs from different and diverse sources, especially body fluids as well as tissues, remains one of the most challenging issues in the EV field as there are very few standardized protocols available. This issue is further complicated by the specification of the research question of the respective scientist, which can lead to deviations. Furthermore, the established protocols partially vary from laboratory to laboratory, even if the same physicochemical principles are used for separation. Position papers by members of the ISEV are therefore of crucial importance to list the requirements and pitfalls when isolating EVs from noncell culture sources. We recommend the readers to refer to Witwer et al. for isolation methods concerning biofluids [44] or to Mateescu and collaborators for RNA isolation from EV samples [39].

### 2.1. Reproducibility and Transparency: The EV-TRACK Database

To improve reproducibility and transparency of EV data, the MISEV guidelines recommend the EV-TRACK knowledge database for data evaluation by the EV-METRIC system [45]. Thanks to this system, journal editors can track the assigned EV-TRACK ID which gives them a comprehensive overview of the data and their quality. Briefly, the evaluation is based on different experimental parameters, including information on the isolation method (e.g., dUC conditions), the quantification and quality of the isolated EVs/particles, submission of TEM images, the detection of EV- and non-EV-enriched proteins, the lysate preparation method, and the antibody specifics if such were used for isolation [45].

#### Ad-EV Research on EV-TRACK

Notwithstanding the great efforts of the ISEV and the current research interest in adipose-derived EVs, reaching 200 publications during the last decade (Figure 1), only 64 entries can be found in the EV-TRACK database when searching for “adipose”, “adipose tissue” or “adipocytes” terms (in combination). These entries belonged to a total of 35 articles, from which only 10 correspond to the study of “adipocyte”-derived EVs. Moreover, only half of the records showed an EV-METRIC higher than 50%, and 12 of them showed a value of 0% (updated in January 2021). These results reflect the lack of standardization in the field and highlight the necessity of further reporting.

Despite the above-mentioned limitations, the combination of EV-TRACK records for adipose-related references and the focus on major concerns addressed by the MISEV guidelines show that the adipose research field follows the usual trends already described for the general EV research (Figure 2). For instance, dUC is the most widely used method for EV isolation (all entries referred to it), followed by the combination of filtration through a 0.22 μm mesh (Figure 2a). Notably, no entries using size-exclusion chromatography (SEC) are reported despite the many benefits described for this technique [34]. Regarding EV origin, cell culture supernatant is the most common material (ca. 74%), although the cell source is often not stated (ca. 55%), and redundant terminology tends to appear (e.g., “adipose-derived mesenchymal stem cells” vs. “adipose mesenchymal stromal cells” vs. “pluripotent stem cells”). Moreover, non-strict adipose tissue material is reported (e.g., “osteoclasts” or “dermal fibroblasts”) within this set of entries. In relation to markers, more than half of entries showed at least one EV marker (Figure 2b), with CD63, CD9, and Tsg101 being the most cited ones (Figure 2c). In contrast, less than 50% reported at least one contamination marker (Figure 2b), underlining the lack of testing of EV purity. The most common non-EV markers were calnexin, Grp74 as well as albumin (Figure 2d). Altogether, these data provide some information for Ad-EV researchers in order to select potential EV/non-EV markers to complement future studies. Nevertheless, the overview of the EV-TRACK database clearly emphasizes the lack of data reports and standardization. These facts should encourage future researchers to increase the spectrum of methodologies applied (i.e., SEC) and markers tested (i.e., co-isolation contaminants) and to report the cell culture conditions in more detail. 

## 3. Considerations When Working with Adipose Cells

Apart from the technical hurdles inherent to any EV research, adipose tissue as question of choice constitutes an additional challenge. This is due to the peculiarities of adipose tissue in terms of depot and its localization, heterogeneity of its cellular composition, plasticity, and the limitations in reproducing these characteristics in vitro (Figure 3). In the next sections, we will depict these peculiarities by referring to previous work in the field and their achieved goals. In addition, special focus will be put on the attempts made on primary adipocytes, which has been hampered by various technical and biological limitations.

### 3.1. Differences between Adipose Tissue Depots

When we talk about fat, we mostly refer to white adipose tissue (WAT). Together with its endocrine role, its main function consists of fat storage (primarily in the form of triglycerides), although it also has an essential role in providing mechanical support as well as thermal isolation to the body [1]. WAT is spread through the whole organism but two main depots can be distinguished: subcutaneous (directly underneath the skin and responsible for the accumulation of around 80% of total WAT) and visceral (which accounts for 10–20% and is mainly constituted by the *omentum* and the mesenteric fat) [46]. In obesity, excessive expansion of WAT leads to adipocyte dysfunction and metabolic disturbance [47]. Of note, not subcutaneous but visceral obesity has been tightly linked to the development of comorbidities, and substantial differences in the morphological, functional, and endocrine levels have been described for both depots (reviewed elsewhere [1,47,48]). Given the differential roles exerted by different WAT depots, one could assume that the released EVs will exert differential molecular profiles and modulatory roles on target cells. Kranendonk and collaborators have performed pioneering studies addressing these questions [49,50]. By using subcutaneous and omental adipose tissue explants, the authors characterized the adipokine profile of EVs and their modulatory functions. In a first study, they relied on sucrose density ultracentrifugation as an isolation method and on a multiplex immunoassay for the proteomic characterization [49]. Thanks to the comparison of total secretome of the separated EV and soluble fractions, the authors confirmed that of six key adipokines were encapsulated in EVs, and their differential concentrations were dependent on the tissue of origin (i.e., visceral depot releases more encapsulated proinflammatory cytokines such as IL-6, MIF or MCP-1). Interestingly, in the same work, the authors also proved the reciprocal signaling between adipocytes and macrophages [49], which was studied in more detail by Flaherthy III et al. [51]. In a second study, Kranendonk et al. also tested the influence of visceral and subcutaneous EVs on liver and muscle cell insulin signaling [50]. In this work, the authors relied solely on dUC as the isolation method and despite that they could confirm a regulatory role of these EVs on hepatocytes, no further conclusions were obtained. Although the authors attributed these contradictory results to the low number of samples analyzed as well as potential discrepancies in the adipokine content [50], the dUC preparation may have contributed to reducing their functionality and influenced their results. Moreover, upcoming studies (Camino et al., currently under review) evaluating the global proteome content of subcutaneous and visceral adipose tissue-derived EVs are cited in this Special Issue [23]. Hence, further studies addressing these differences will increase our knowledge on the WAT endocrine role by extending the range of functions exerted by both depots.

Apart from subcutaneous and visceral fat, secondary WAT depots in the organism are also found in the bone marrow, neck, skeletal muscle, liver, and around the blood vessels (perivascular adipose tissue) or the epicardium (epicardial adipose tissue). Interesting work from Li et al. has recently shown that perivascular adipose tissue-derived EVs mediate vascular remodeling through miR-221-3p in the context of obesity-mediated inflammation [52]. To achieve these results, the authors analyzed EVs released by mesenteric adipose tissue explants from mice as well as in vitro-differentiated preadipocytes (primary as well as 3T3-L1 cells) isolated by dUC. Further purification steps or size characterization of EVs were not performed. However, the authors evaluated the expression of several EV-associated protein markers by Western blot and additional EV trafficking and uptake assays were shown [52]. These findings underlined the paracrine regulatory role of WAT not only by adipokine release but also by EV-encapsulated microRNAs. In addition, recent studies have pointed out the enrichment of mRNA and noncoding RNAs in the total secretome of epicardial adipose tissue [53], which guarantees future investigation focusing on these “secondary” depots.

#### Brown Adipose Tissue (BAT)

There is another type of fat depot known as brown adipose tissue (BAT). The presence of BAT had long been ascribed to newborns and mammals with hibernation capacity, given its thermogenic function [54]. However, in 2009, the presence of active BAT was also characterized in adult humans [55,56], leading to an exponential increase in research interest about its metabolic regulatory role. BAT and WAT are distinct in their molecular, morphological, and functional characteristics as well as their embryological origins [1,57]. BAT is specialized in heat production upon cold exposure; its brown adipocytes are much smaller and mitochondria-enriched compared to the white ones. Fat storage occurs in multilocular small droplets, and the presence of other accompanying cell types in the tissue is scarce. In the context of obesity, the maintenance of BAT depots in mammal adults and/or the process of *browning* of white fat has been associated with better metabolic performance and a decrease in the development of comorbidities [58,59]. The research interest in BAT has also become patent in the EV field, where we can already find a few articles working on brown Ad-EVs [60,61,62]. Among other interesting results, brown Ad-EVs have been shown to mitigate metabolic syndrome in a high-fat diet mouse model [60] and to regulate hepatic lipogenesis through the exosomal miR-132-3p [61]. In addition, Chen et al. validated the brown-exosomal origin of miR-92a whose circulation levels are inversely correlated with human BAT activity [62]. Although BAT-derived EVs are of utmost importance [63], the number of publications on this topic is still limited. Therefore, in this manuscript, we have focused on the WAT-related references, which are referred to as “adipose tissue” for simplicity.

### 3.2. The Heterogeneity of Adipose Tissue Composition

Adipose tissue comprises adipocytes in addition to a wide population of cells known as the stromal-vascular fraction (SVF). The heterogeneity of SVF relies on the diversity of its cells which include macrophages, fibroblasts, blood cells, endothelial cells, smooth muscle cells, lymphocytes, mesenchymal stem cells (MSCs), and adipose precursor cells. The phenotype and cell composition of the SVF vary according to the body depot, the adiposity, as well as the physiopathological state of the tissue [1,47]. Since SVF considerably contributes to the adipose tissue proteome [64] and secretome [65,66], these differences will also impact the EV population, introducing potential bias when comparing EVs from adipose tissue explants to EVs coming from mature adipocytes or any other specific cell type (e.g., macrophages or endothelial cells). As already mentioned, there are references of interest in the context of obesity and diabetes comparing the molecular and functional profiles of EVs released by total adipose tissue explants [49,50,67], as well as others working specifically with EVs from visceral samples [68]. Similar works can also be found for mice adipose tissue explants [27,51,69]. However, specific contributions of the adipose tissue components to the EV population are mostly unknown and may depict new functional capabilities of each particular cell type and their contribution to the total adipose tissue secretome. 

#### 3.2.1. Primary Mature Adipocytes: Ex Vivo Culture for Ad-EV Research

Adipocytes are the main component of adipose tissue, and therefore the study of these primary cells will give us a snapshot of the physiological state of this tissue and its potential regulatory capabilities. Despite decades of investigation, the isolation and maintenance of primary mature adipocytes still constitute a major challenge leading to a lack of translational implementation of in vitro/ex vivo models [70]. Adipocyte isolation requires enzymatic digestion of the tissue combined with mild centrifugation for their separation from the SVF [71]. However, adipocytes are big cells with high lipid content which make them float and thus easy to harvest; however, they also tend to break, which hampers a good yield as well as limits their culture. It should be noted that mature adipocytes do not grow or expand under “culture conditions” but are just preserved. In this regard, the adipogenic and senescent phenotype of cells must be tested over time [70,72], which also applies to the EV field and the timing of conditioned media collection. Due to the potential of lipid-based artefacts (i.e., lipid droplets or adipocyte membrane debris), additional controls such as time-point-zero samples are highly advisable. 

Thanks to the natural buoyant properties of freshly isolated mature adipocytes, these cells are usually maintained in cell culture media as a “floating” system. In this regard, EV release will occur in the “infranatant”, which supports the nonadherent floating cells. An additional method working with primary adipocytes is the ceiling culture, which consists of allowing the floating cells to contact the ceiling of a culture flask by filling it completely with medium and waiting for the cells to attach [73,74]. However, several works have addressed how adipocytes maintained under these conditions rapidly dedifferentiate to fibroblast-like cells [75], entailing a suboptimal approach. An additional interesting method to be applied to Ad-EV research is the membrane mature adipocyte aggregate culture (MAAC) method, recently described [70]. Remarkably, Harms et al. have demonstrated that MAAC is a versatile tool for studying phenotypic changes of mature primary adipocytes providing an improved translational model. In their work, MAAC and short-term floating cultures were revealed as the best options to preserve mature adipocyte identity and function [70]. A major limitation of MAAC is the cost of the membranes used in the approach (transwell inserts and plates); although, at the same time, this system allows coculture with other cells (e.g., macrophages), which is highly desirable. Therefore, future studies on Ad-EVs obtained under the MAAC system are likely.

To our knowledge, few references have assessed primary mature Ad-EVs despite the relevance of this cell type in metabolic homeostasis. Most of the Ad-EV references up to now have relied on in vitro-differentiated adipocytes from primary material as well as immortalized cell models such as 3T3-L1 (see next sections and reviewed elsewhere [22,23]). Although scarce, references working with freshly isolated cells usually provide well-described methods but culture conditions still lack standardization and supplementary controls (e.g., tracking of EVs released by potentially contaminating SVF). Given the intrinsic difficulties entailed by mature and floating adipocytes, a systematic and exhaustive description of materials and methods in future research is advisable. A summary of the current most relevant references is provided in Table 1. Of note, only two of these works have been currently submitted to the EV-TRACK repository despite their relevance in the field.

To the best of our knowledge, the first article to evaluate primary mature Ad-EVs was published by Müller et al. [24]. In this article, the authors assessed the transcriptional molecular profiles (mRNA and microRNA) of EVs coated by GPI-anchored proteins (Gce1 and CD73), released by small and large rat adipocytes. By using different isolation and purification methods, the authors confirmed the transfer of EV-encapsulated transcripts from large to small adipocytes, highlighting their paracrine regulatory role in lipid synthesis. Later, Lee et al. extended these results to the proteomic level [25]. In this work, the authors identified more than 500 proteins associated with primary rat Ad-EVs. Although additional purification methods for EV samples (isolated by dUC) were not described, particle sizing was estimated by NTA, and confirmation of EV presence was carried out by TEM. The same year, Eguchi et al. described how Ad-EVs promote macrophage migration [26]. Despite the interesting conclusions of the work, most of the approaches described were focused on 3T3-L1 in vitro-differentiated adipocytes and circulating EVs from adipose tissue origin, and no conclusions for primary cells in terms of technical resonance can be extracted. In contrast, Durcin and colleagues published a work precisely detailing the origin of the characterized EVs some years later [29]. In this work, the authors described notable differences between large and small EV subpopulations obtained by dUC through a proteomics approach. Furthermore, the majority of the experiments were performed by studying EVs derived from 3T3-L1 in vitro-differentiated adipocytes. However, primary murine Ad-EVs were used for validation [29], confirming the presence of CD9, CD63, and flotillin-2 among other markers in these samples.

Of note, Lazar and coworkers were the first group to study human primary Ad-EVs [27] in the context of cancer research. By using a combination of results from in vitro 3T3-F442A-differentiated as well as mice and human mature adipocytes, the authors demonstrated that Ad-EVs promote the aggressiveness of melanoma cancer cells, which in turn is aggravated under obesity conditions. Among other results, it is especially noteworthy that the authors showed a positive correlation of human Ad-EV release to BMI (characterized by NTA), and how “obese” vesicles increased the migration capacity of melanoma cells [27]. Additionally, the validation of some EV markers such as flotillin-1 was also shown. Moreover, the same group has recently confirmed the role of Ad-EVs in fueling melanoma tumors [30]. Of note, the different approaches presented by Clement et al. represent one of the best-described articles on Ad-EVs research to date. In this work, the authors extend the conclusions of their previous work [27] by proteomics approaches on murine primary Ad-EVs as well as a valuable set of functional assays including EV dosage, fatty acid content quantification, and trafficking through confocal microscopy, among others [30]. Outstandingly, to confirm that EVs were responsible for fatty acid transfer and not other structures potentially co-isolated by dUC such as lipoparticles, the analysis of SEC fractions confirmed that only fractions containing EVs could cause lipid droplet accumulation in melanoma cells as well as exert a promigratory effect [30]. Additionally, how cancer-associated adipocytes (CAAs) modulate chemoresistance in ovarian cancer (OC) through miR-21a by suppression of apoptosis has been also shown [28]. In this work, the number of human samples (healthy and cancer-associated) was limited (n = 2) and EVs were not further purified by additional methods; however, the authors showed complementary results including EV trafficking analyses on OC cells as well as the characterization of the Ad-EV transcriptome by next-generation sequencing (NGS). Although the authors relied on ceiling culture for the primary adipocytes, NGS results showed distinctive clusters for the EVs derived from OC cells, fibroblasts, and adipocytes [28], which underlines the significance of their results. 

The references described in this section constitute pioneering work on Ad-EVs, especially considering the intrinsic complications of handling adipose tissue and primary adipocyte isolation. dUC, which has been extended to the whole EV research field, constitutes the most widely used method to study these particles so far. However, additional efforts made to achieve EV purification such as density gradient or SEC methods have notably increased the impact of the results obtained [30], which are the focus of an additional review in this Special Issue [34]. The implementation of newly developed, state-of-the-art methodologies aimed to characterize EV size, surface markers as well as imaging will help to increase our knowledge not only on the obesity-cancer binomial, but also on any context in which adipose tissue exerts its natural endocrine role.

#### 3.2.2. Adipose-Derived Stem Cells (ASCs)

One cellular compartment of the SVF receiving special attention in the EV field is the adipose tissue MSCs. These cells are referred to by different names (such as ASCs, AdMSCs, or ADSCs) (nicely reviewed by Ruiz-Ojeda et al. [76]), so, according to the latest consensus, herein we will refer to them as “adipose-derived stem cells” (ASCs) to identify easily accessible, plastic adherent, multipotent stem cells. Of note, subcutaneous in contrast to visceral fat has been observed to be enriched in ASCs [71], most probably responding to the better expandability of this tissue [77]. Due to their adherent nature, ASC-associated EV research entails common advantages and disadvantages of all in vitro cultures, where EV isolation is based on conditioned media collection and processing [34]. The well-known capabilities of these cells are their adipogenic, chondrogenic, osteogenic, and myogenic differentiation properties [78]. Additionally, ASC effects on cancer [79] as well as cardiovascular and neurodegenerative diseases [80] have been described, underlining their potential use in regenerative medicine [81]. Due to these facts, the characterization, as well as the potential function of EVs derived from multipotent ASC-EVs or ASC-derived adipocytes, have been extensively revised [19,82,83]. Despite these promising alternatives, it should be emphasized that isolation and culture conditions of ASCs can have a significant and rapid impact on ASCs’ secretome [84]. Moreover, the ASC differentiation grade to mature adipocytes could also vary due to several factors [81], including but not limited to the culture conditions and the (patho-)physiological conditions of the tissue of origin. These outcomes will undoubtedly impact EV composition, which again underlines the necessity of standardization and superior description of experimental conditions in future studies. 

#### 3.2.3. Other Remarkable Cells from the Stromal-Vascular Fraction (SVF): Current Knowledge

Although still scarce, we can find a few references analyzing specific cell types from the SVF such as adipose tissue macrophages (ATMs) [85], fibroblasts [28], or endothelial cells [86]. More concretely, it has been demonstrated that EVs derived from ATMs can regulate insulin signaling on adipocytes, myocytes, and hepatocytes in vitro and in vivo through miR-29a [85]. As mentioned above, it has been shown that omental adipocytes and fibroblasts can transfer EV-encapsulated miR-21 to OC cells, causing resistance to taxane-based chemotherapy [28]. Interestingly, Au Yeung et al. also showed that EV release was significantly higher in cancer-associated fibroblasts (CAFs) compared to CAAs, according to the tunable resistive pulse sensing (TRPS) quantification [28]. These results pointed out a differential contribution of both cell types to the tumor secretome. Notably, in a keynote work from Scherer’s group [86], the potential bias between different adipose tissue components was also evidenced. Intriguingly, the direct exchange of protein and lipid signals between endothelial cells and adipocytes through EVs was discovered during the attempts of generating an adipocyte-specific knockout mouse model of caveolin 1 (cav1) without success. Despite the adipocyte-specific elimination of cav1 in vivo, the authors still observed cav1 expression in these cells. Thanks to the primary culture of endothelial cells and subsequent EV isolation by dUC and sucrose gradient purification, the authors determined that endothelial cells transfer cav1-containing EVs to adipocytes in vivo and that this phenomenon is regulated by the nutritional state of the animal [86].

The study of the separated fractions constituting the adipose tissue has helped us to deepen knowledge of their functional capabilities as well as to improve our understanding of the inherent heterogeneity of the EV population. Nevertheless, one should be aware that working with adipose tissue explants will offer a more physiological and closer view of the in vivo situation. This is particularly significant since the 3D structure of adipose tissue has been observed to determine its inflammatory state [87,88] as well as its adipogenic potential [89,90], which in turn may impact the EV molecular profile. To achieve this aim, new 3D systems are being designed for the in vitro evaluation of adipocytes alone or together with other cell types (e.g., macrophages) (reviewed in [76,81]). On the downside, it has been described that ex vivo culture of adipose tissue explants is associated with a rapid loss of the phenotype determined by inflammation and hypoxia [70,91,92], which may also impact EV release and/or profile. Therefore, the implementation of 3D culture approaches in combination with EV isolation, characterization, and tracking methods will also be of utmost importance in future studies.

### 3.3. The Adipogenesis Process

Part of the dynamism of adipose tissue lies in its expansion capacity. This expansion can occur through an increase in adipocyte volume (hypertrophy) and/or number (hyperplasia), which in turn requires suitable extracellular matrix remodeling [93,94]. Adipocyte differentiation or adipogenesis is a complex and tightly regulated process in which several key steps could be distinguished: (i) adipocyte progenitor recruitment from the SVF; (ii) mitotic clonal expansion of preadipocytes and signaling cascade induction involving transcriptional factors such as cyclic AMP response element-binding protein (CREB), CCAAT/enhancer-binding protein (C/EBPβ) and peroxisome proliferator-activated receptor-γ (PPARγ); (iii) expression of genes determining adipocyte fate such as FABP4, fatty acid synthase (FASN), adiponectin (ADIPOQ) or glucose transporter 4 (GLUT-4), among others [95,96,97]. Ultimately, the expressions of these adipogenic genes will allow the accumulation of triglycerides in lipid droplets, therefore contributing to the enlargement of these cells. 

In the last few years, different cell culture models and protocols have become available to study adipogenesis and adipocyte biology, as previously revised [76,81]. We recommend the readers refer to these references in order to learn more about the limitations when working with these cells. Major concerns include (but are not limited to): in vitro-differentiated adipocytes are dramatically smaller in size and never display the unilocular in vivo appearance; in vitro differentiation is a complex process that requires the addition of an artificial hormonal cocktail; and differentiation rate may be affected by several factors such as cell confluence, passage number or serum lot [98]. Nevertheless, previous work on these models has undoubtedly increased our knowledge of Ad-EVs and should not be underestimated. Of note, the work from Conolly and collaborators showed how the adipogenesis process determines Ad-EV release and composition, by working on 3T3-L1 cells [99]. In this work, preadipocyte-derived and mature Ad-EVs were recovered and analyzed in terms of size (by NTA), lipid composition (by gas chromatography with flame ionization detection), and annexin V positivity (by flow cytometry). In addition, an interesting approach combining ELISA type detection with permeabilization allowed for the detection of adipocyte-related markers such as FABP4 or PPARγ in the aforementioned vesicles. Interestingly, the authors concluded that preadipocytes have a higher EV release rate and the lipid composition of EVs is determined by the adipogenic state of the cell [99]. Of note, this article pointed out the importance of well-suited in vitro models and established protocols as well as a detailed description of experimental conditions. Thus, differences in the in vitro adipogenic rate as well as different time-points for EV collection may explain potential discrepancies in lab-to-lab reproducibility, which is an extended shortcoming in the EV field [16].

Other works already detailed in this manuscript have also investigated Ad-EVs’ functional role by using 3T3-L1 and 3T3-F442A in vitro-differentiated adipocytes, further validating their results with primary ex vivo Ad-EVs [27,29,30]. Recent noteworthy results are the phenotype switch suffered by Ad-EVs released by pre-adipocytes and mature adipocytes under insulin-resistant and hypertrophic conditions [31]. By a high-throughput proteomic approach working on C3H10T1/2 cells, Camino et al. demonstrated how Ad-EVs reflect the pathological state of the cell of origin and can exert a worsening of the phenotype in a healthy recipient (in this case, an insulin-resistant phenotype). Interestingly, these modulatory effects were demonstrated for adipocytes but also macrophages as recipient cells [31], confirming previous work on the reciprocal signaling between these two cell types [49,100]. Moreover, this article has a detailed technical description of the culture and EV handling (i.e., volume collection media, preincubation times, vesicle:cell ratios, etc.) and also performed several orthogonal approaches for EV characterization (i.e., NTA, Western blot, TEM). Outstandingly, this article was the first to provide additional mechanistic insights about the paracrine role of Ad-EVs on adipocytes, which have been shown to regulate the adipogenic differentiation state through Akt phosphorylation [31].

## 4. Downstream Applications Using Ad-EVs

As already highlighted in previous sections, the researchers have to adapt their methodological approach considering the subsequent analysis to be carried out. Although this assertion may seem trivial, recent studies have underlined how usual praxis in the lab could affect EV preparations and, therefore, interfere and impact the results. For example, several consecutive ultracentrifugation steps could affect the morphology of EV samples, hampering their potential functional effects on target cells [101] as well as freeze and thaw cycles, which have been observed to dramatically reduce the number of particles compared to freshly isolated samples [102]. Moreover, the addition of low concentrations of mild detergents in buffer solutions in addition to short-time incubations at room temperature has been observed to improve EV collection and suspension during isolation procedures in our lab. Again, these facts underline not only the importance of methodological standardization in the field but also the necessity of carefully detailed procedures, especially when new and innovative approaches arise. 

### 4.1. Considerations for Transcriptomics

The discovery of genetic material in EVs has become evident in recent years through various techniques, thus extending the functional spectrum of the individual EV subpopulations. With the advent of high-throughput RNA sequencing methods, different RNA species have been isolated in subpopulations of EVs, of different biological origins, and from different cell culture conditions. These RNA species mainly include noncoding RNAs such as miRNAs, lncRNAs, snoRNAs, tRNAs, rRNAs, piRNAs, Y RNAs, and circRNAs [103,104,105,106,107]. In addition, fragments as well as functionally active mRNAs have also been detected in EVs [108]. Recently, a conflicting hypothesis has emerged stating that RNAs, in particular the widely studied miRNAs, are less commonly found within EVs than initially assumed. Rather, miRNAs appear to be found as stable extracellular RNAs in non-vesicular fractions [109]. Hence, when analyzing Ad-EVs (as with EVs from other sources) the same prudence and consistent strategies should be used to exclude any possible misinterpretation. As stated above, the lack of standardization concerning EV isolation, which influences the heterogeneity of the EV population, also affects the degree of contamination with non-EV-associated RNAs of different origins [110]. In addition to the purification method of EVs, the chosen RNA isolation technique can also lead to different results, especially when specific RNA classes are intended to be analyzed [111,112]. Since the amount of RNA from EV material is often a limiting factor, the quality control should be adapted accordingly. The same concerns apply to the choice of library preparation and the selection of the sequencing platform. A compendium of the individual points to be considered can be found in a position paper of the ISEV [39].

### 4.2. Considerations for Proteomics

The characterization of specific markers constitutes one of the major challenges in the EV field [16,113]. In addition, EV function largely depends on the molecular cargo, and surface composition may be vital to exert a functional effect on the recipient cell. State-of-the-art high-throughput proteomics approaches are capable of identifying and detecting subtle changes in protein levels even for low abundance protein species [114], which make them an anticipated tool for EV research. Notably, the bulk of data generated to date has allowed the depiction of organelle maps by proteomic profiling, also covering the endosomal compartment (see [115] for a review). Far from classic *label-free* approaches, advances in relatively new labeling methods (i.e., TMT, iTRAQ, or SILAC) [116] and the application of robust quantification algorithms are also highly recommendable [117]. However, in contrast to other cell types or tissues [118], the adipose-derived EV composition is still being deciphered and only a few studies have relied on proteomics approaches for the analysis of primary material from human [17,49,68] or animal origin [25,119,120]. 

One key feature of mass spectrometry is the high-throughput identification and quantification of a plethora of proteins. However, the interindividual heterogeneity of primary material, especially when dealing with human specimens [121], makes the interpretation of data difficult. Depending on the major aim (hypothesis-free vs. hypothesis-driven purpose), sample pooling prior to proteomics becomes a reasonable alternative, and several publications have illustrated how pooling samples could counteract this heterogeneity [122,123]. Nevertheless, we want to point out that when working with pooled samples it is of the utmost importance to mix exactly the same amount of protein from each individual. In this regard, we will only recommend pooling purified EV samples and/or performing this approach based on the particle concentration ratio. To our knowledge references for EV proteomics based on particle number have not been described so far due to the fact that EVs have mostly been isolated by dUC, which normally entails protein co-isolation (allowing the quantification of those proteins). Thus, technical investigation on EV-particle number-based approximations will be also interesting in the near future. 

A review on proteomics studies on adipose tissue-derived EVs by Camino et al. [23] is also part of this Special Issue. Since a fully detailed description of Ad-EV markers goes beyond the scope of our article, we strongly recommend the readers to read their work. Based on their investigation, the authors highlighted PLIN1 as a marker abundantly present in Ad-EVs, together with others such as cystatin C, FABP4, mimecan, or TFBI. In addition, another contribution to this Special Issue has also revised the Ad-EV proteomic composition in the context of obesity [43]. The combination of these markers would allow the confirmation of adipose tissue-origin of EVs, especially when working with biofluids [23]. For reviews on adipose-derived circulating EVs, we refer the readers to Camino et al.’s work on proteomics [23] as well as to Mori et al.’s for microRNA-focused research [124].

### 4.3. Considerations for Functional Studies

There is a general risk of overinterpretation and artefact detection when using EVs in functional studies and thus careful consideration of adequate controls needs to be taken. One thing to consider is the method of cultivation of EV-producing cells and subsequent purification of EVs. For example, EV composition is dependent on cell homeostasis; cells under hypoxic stress release more EVs than under normoxic conditions, and EV size and cargo are altered [125]. As previously mentioned, adipose tissue explants are especially prone to hypoxic stress [92], which may alter the EV phenotype and release.

When isolating EVs, the method of choice must be carefully evaluated. Commercial kits often isolate EV types with specific markers while dUC isolation harvests indiscriminately but also pelletizes non-EV structures such as protein aggregates. Similar co-isolation problems are also a handicap when using precipitation-based kits. Furthermore, the high centrifugal forces during dUC alter the integrity of EVs and likely influence their biological function [101,126]. With commercial kits, unwanted assay components such as polymers, antibodies, beads, and solvents are introduced to the functional assay which makes interpretation of data difficult. Moreover, isolation of specific subgroups may eliminate the most active EV subtype for a particular functional assay. Of critical importance for the analysis of EV-associated and EV-independent biological activities are dose–response studies that include negative controls. Additional alternatives could be conditioned medium as a background control compared to complete medium without conditioning by cells which are processed the same way as the conditioned sample. Furthermore, controls could include samples in which EVs were eliminated beforehand. In this context, no standards are available yet, but some controls or EV dosages have been listed in Ad-EV publications [30,31,49,51,127].

Especially challenging are those studies concerning biofluids, in which a comparison of healthy vs. pathological fluids (i.e., disease-free, matched donors) is highly advisable. In order to exclude effects from macromolecular non-EV components, methods such as density gradients or SEC are advised as they improve the separation of EVs from non-EV components. The different fractions should be evaluated in preliminary experiments of the functional assay to exclude unspecific effects. Similarly, EV depletion controls should lead to loss of activity in the assay and should be at least performed in preliminary experiments. Although still limited, some Ad-EV publications have addressed these concerns [24,27,30,31]. 

## 5. Challenges and New Perspectives

Despite the great advances in EV research during the last decade, there are still many challenges in the field (reviewed in [113,128]), which are certainly extensible to the Ad-EV research.

First, there is still a lack of specific EV markers as well as standardization of adequate isolation methods depending on the purpose of the study (although, as previously mentioned, several groups and initiatives are working hard on these issues [14,15,16]). In the case of Ad-EVs, despite specific markers from adipocytes are guaranteed, most of them are based on intracellular molecules limiting the spectrum of methods to be applied and, therefore, the conclusions of the study (e.g., nFCM relies on surface extravesicular markers). In addition, it should also be noted that specific extracellular markers from white, beige, or brown adipocytes are scarce [129,130], and most probably not all EVs will carry those markers as reflected by the innate heterogeneity of the EV population [15], which has also been described for Ad-EVs [29]. In this regard, future studies with highly purified Ad-EV samples in combination with high-throughput proteomics and state-of-the-art EV characterization techniques will overcome these limitations by revealing new and/or highly abundant adipocyte extracellular markers.

Second, well-established methods (i.e., dUC) have proven to be disadvantageous, especially when it comes to functional studies and modulatory effects of EVs. Differential lipid composition from Ad-EVs or any other EV type may also influence the negative impact of different approaches and experimental conditions. For instance, dUC could affect EVs according to their plasticity: the more “flexible” ones may be less affected than the more “rigid” ones and therefore retain their original capabilities better. Interestingly, differential stickiness of EVs has been supported in the context of cancer [131] and ongoing experiments in our lab have evidenced technical hurdles in EVs from different origins in this regard (*data not shown*). Thus, the evaluation of other biological parameters such as mechanical forces turned out to be relevant for the understanding of EV functional capabilities.

Third, the functional significance of EVs remains in question due to the persistent perception that they are inactive, and physiologically irrelevant shed vesicles [128]. Although their modulatory action on target cells has been repeatedly demonstrated, their physiological relevance is still under debate. It should be noted, however, that EVs need to be regarded as biological entities by which the sole contact to the target cell could result in a signaling cascade which, in turn, could be reinforced by the action of a single and/or a group of transferred encapsulated molecules. New methodologies and improvements on the targeting of EVs may help to shed light on this matter.

In conclusion, Ad-EV research during the last decade has provided evidence that adipose tissue communicates with other cell types via EVs. In addition, a set of markers (such as CD63, PLIN1, or FABP4) have emerged as promising tools for Ad-EV evaluation and detection. Thus, the translational impact of Ad-EVs as therapeutic tools and/or targets is expected. Groundbreaking studies addressing the potential clinical relevance of Ad-EVs have been already published (reviewed in [43]). For instance, Ad-EVs have been shown to stimulate mitochondria supporting melanoma progression [30] and obese adipose-derived EVs contribute to the development of insulin resistance [100]. Despite these facts, the lack of standardization in EV methodology and limited protocol information in published data still hamper the conclusions achieved up to now. The absence of specific Ad-EV markers also limits the targeting of these particles for clinical purposes, especially when considering that current markers have been described to be modulated under pathological conditions (i.e., FABP4 and obesity) or are a currently a matter of concern due to their potential co-isolation as non-EV structures (i.e., adiponectin or PLIN-1) [43]. Future efforts in this regard will undoubtedly improve our knowledge of Ad-EVs characteristics and potential therapeutic approaches and guarantee a better understanding of the mechanisms behind the homeostatic role of adipose tissue. 

## Figures and Tables

**Figure 1 ijms-22-03362-f001:**
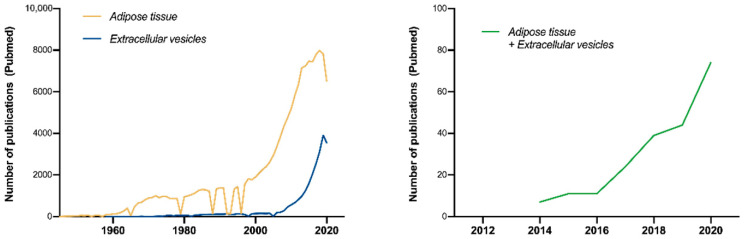
Timeline for “adipose tissue” and “extracellular vesicles” publications. Lines show the number of publications per year corresponding to the keywords “adipose tissue” (dark yellow, left panel), “extracellular vesicles” (blue, left panel), or a combination of both (green, right panel) in the PubMed database (https://www.ncbi.nlm.nih.gov/pubmed, accessed on 31 December 2020). Note that both terms follow a similar trend. The slight decrease in the number of publications in 2020 for both individual topics may be due to the pandemic situation caused by SARS-CoV-2 infections [20,21].

**Figure 2 ijms-22-03362-f002:**
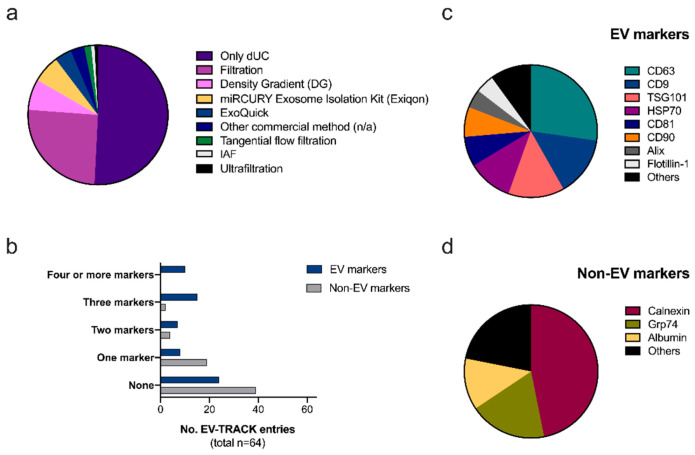
Overview of EV-TRACK adipose-related entries. Records for “adipose”, “adipose tissue” and “adipocyte” search terms on the EV-TRACK knowledge database were combined and analyzed. A total of 64 entries were retrieved by the 31 December 2020 (https://evtrack.org/) [45]. (**a**) Pie chart showing the relative proportion of isolation methods used. It should be noted that differential ultracentrifugation (dUC) was present in all entries; therefore, the rest of the methods reported were applied (if so) in combination. (**b**) Bar graph reflecting the number of entries reporting none, one, two, three, four, or more extracellular vesicle (EV) (blue) and non-EV (grey) markers. Pie charts reflect the relative proportion of EV (**c**) and non-EV (**d**) markers reported in the different studies (single or in combination with others).

**Figure 3 ijms-22-03362-f003:**
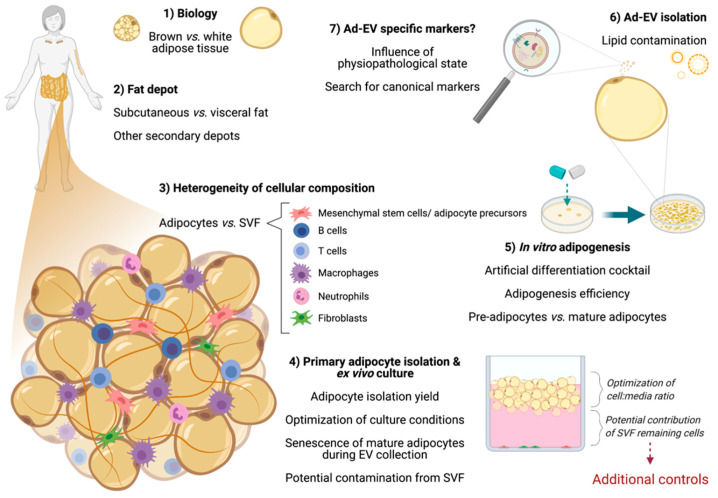
Biological and technical aspects to consider when working with adipose cells. Schematic overview of principal aspects related to white adipose tissue biology and the methodological approaches for its study. Derived challenges for each point are also addressed. Most of the studies performed up to date are based on adipose tissue explants (implying a heterogeneous population of derived extracellular vesicles) or adipogenesis in vitro models. Primary adipocyte ex vivo culture conditions are still a current challenge (due to the lack of standardization) and additional controls in future studies are highly advisable (especially regarding the potential contamination of the stromal-vascular fraction (SVF)). Lipid contamination during adipocyte-derived (Ad)-EV isolation has recently been shown as a potential drawback. The absence of specific Ad-EV markers remains an obstacle in the field. Ad-EV, adipocyte-derived extracellular vesicles; SVF, stromal-vascular fraction. This figure was created with BioRender.com (accessed on 16 March 2020).

**Table 1 ijms-22-03362-t001:** Overview of currently published references working on Ad-EVs obtained from primary isolated adipocytes.

Reference	Müller G et al., 2011 [24]	Lee, JG et al., 2015 [25]	Eguchi A et al., 2015 [26]	Lazar et al., 2016 [27]	Au Yeung et al., 2016 [28]	Durcin et al., 2017 [29]	Clement E et al., 2020 [30]
EV-Track No. (EV-METRIC)	EV110050 (43%)	*N/A*	*N/A*	*N/A*	EV210034 (45%)	*N/A*	*N/A*
Material source	Primary rat adipocytes (male Sprague–Dawley or Wistar)	Primary rat adipocytes (male LETO and OLETF)	Primary mice adipocytes (epididymal AT of *ob/ob* mice)	Primary mice and human adipocytes (subcutaneous)	Primary human adipocytes (omentum)	Primary mice adipocytes	Primary mice and human adipocytes (subcutaneous)
Conditioned media (composition)	No specific depletion described	No specific depletion described	No specific depletion described	EV-depleted (ON) media	Medium with exosome-free FBS	Serum-free media	EV-depleted (ON) media
Adipocyte primary culture/incubation	Microfuge tubes prefilled with dinonylphtalate	Ceiling culture (preincubation not indicated)	*N/A*	Floating culture	Ceiling culture (for 5–7 days)	Floating culture	Floating culture
Conditioned media (collection time)	2 h	3 days, every 24 h	40 h (?)	24 h	48 h	24 h	24 h
EV isolation method	dUC + Sucrose density gradient + Affinity purification	dUC + Filtration (0.22 ∅ μm)	dUC	dUC	dUC	dUC	dUC
EV sizing characterization	N/A	NTA and TEM	DLS and TEM (but only described for in vitro-differentiated cells and plasma vesicles)	NTA * and TEM	TRPS * and TEM	NTA * and TEM	NTA * and TEM
Primary Ad-EVs markers	FSP27, perilipin-1, CD73, caveolin-1, leptin and others (transcript level)	AQP7, caveolin, CD63, LPL and others (WB)	Annexin V (Flow cytometry)	ECHA, HCDH, FLOT1 (WB)	CD63, HSP70 (WB)	Caveolin-1, CD9, CD63, flotilin-2, Mfge8	ECHA, HCDH, FLOT1 (WB)
Orthogonal experiments	RT-qPCR, SEC	LC/MS, WB	Additional experiments performed on in vitro-differentiated adipocytes	Additional experiments performed on in vitro-differentiated adipocytes (sucrose density gradient, LC/MS)	Ad-EV tracking on ovarian cancer cells	LC/MS, sucrose density gradient, WB	LC/MS, SEC, functional EV tracking

(*) indicates that particle concentration is also described; (?) means that description is unclear. Abbreviations: Ad-EVs, adipocyte-derived extracellular vesicles; AT, adipose tissue; DLS, dynamic light scattering; dUC, differential ultracentrifugation; EV, extracellular vesicle; N/A, not available; NTA, nanoparticle tracking analysis; LC/MS, liquid-chromatography–mass-spectrometry; RT-qPCR, real-time quantitative polymerase chain reaction; SEC, size-exclusion chromatography; TEM, transmission electron microscopy; TRPS, tunable resistive pulse sensing; WB, Western blot.

## Data Availability

No new data were created or analyzed in this study. Data sharing is not applicable to this article.

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
