# Peer review of "Beyond the Extracellular Vesicles: Technical Hurdles, Achieved Goals and Current Challenges When Working on Adipose Cells"

_ijms, 2021, doi:10.3390/ijms22073362_

Round 1
Reviewer 1 Report
Very well written review on a topic that deserve attention.
Author Response
Response to Reviewer 1 comments
General comments: Very well written review on a topic that deserve attention.
Response: We appreciate the positive feedback from the reviewer. As suggested, we have checked the English language and style and performed minor changes as reflected (in red) in the revised version of the manuscript.

Reviewer 2 Report
This is an informative and well written review
Specific points
- Figure 1. The visibility of the yellow line on the left could be improve with a different color. This reviewer cannot see anything on the right panel.
- Can the authors give a concluding paragraph that could help the reader distinguish between extracellular vesicle that might have physiological relevance versus those that are artifacts of the preparation of the adipocytes or adipose tissue? Is there a greater danger of artefacts from adipose tissue derived from obese animals or people where adipocytes are larger and more fragile?
Author Response
Response to Reviewer 2 comments
General comments: This is an informative and well written review
Response: We appreciate the positive feedback from the reviewer and the points addressed which have contributed to increase the quality of our manuscript. As also suggested, we have checked the English language and style and performed minor changes as reflected (in red) in the revised version of the manuscript.
Point 1: Figure 1. The visibility of the yellow line on the left could be improve with a different color. This reviewer cannot see anything on the right panel.
Response 1: We have submitted a new version for Figure 1 with a darker color for left panel in order to improve the visibility as suggested. We apologized for the missing panel in the previous version of this manuscript. This fact may respond to compatibility problems between MacOS and Windows Microsoft Office versions (we have experienced similar problems before). We attach separately here the new figure for the Reviewer´s information and we hope the TIFF version works properly now. In addition, we have checked figure 2 and attached a revised version for panel a in which the size of letters has been increased.
Point 2: Can the authors give a concluding paragraph that could help the reader distinguish between extracellular vesicle that might have physiological relevance versus those that are artifacts of the preparation of the adipocytes or adipose tissue? Is there a greater danger of artefacts from adipose tissue derived from obese animals or people where adipocytes are larger and more fragile?
Response 2: We appreciate the Reviewer´s comments and questions regarding artifactual aspects in Ad-EV research. We have included several comments on potential artifacts from these samples in the revised version of this manuscript (lines 125-126, 154-156, 360-361, 385-386) and a concluding remark (lines 719-723).
Regarding the second question about a greater number of artifacts coming from obese and/or larger size adipocyte samples, we speculate that the potential co-isolation of, for example, lipid droplet artifacts may be increase. However, no references up to now have addressed these questions. We have reflected these concerns in the revised version of the manuscript and additionally referred the readers to a recent review (new reference 43, also part of this Special Issue) which has also highlighted these aspects (lines 154-156).

Reviewer 3 Report
The review Beyond the extracellular vesicles: technical hurdles, achieved
3 goals and current challenges when working on adipose cells is interesting one, but needs more work to improve the content.
The EVs introduction is not new, and the author written many pages.
The importance of standardization section is good one.
Reproducibility and transparency: the EV-TRACK database also good.
Table 1 is good and I will suggest to author if possible please include one more table about the clinical translational aspect of EVs.
Also please include some more figures to highlight the topic more representiative and appealing.
The title is sounds more good than manuscript text, please write comprehensively to streamline the manuscript
Author Response
Response to Reviewer 3 comments
General comments: The review Beyond the extracellular vesicles: technical hurdles, achieved
3 goals and current challenges when working on adipose cells is interesting one, but needs more work to improve the content.
Response: We appreciate the feedback from the reviewer and we hope that the revised version of the manuscript including his/her considerations is now suitable for publication. As also suggested, we have checked the English language and style and performed minor changes as reflected (also in red) in the revised version of the manuscript.
Point 1: The EVs introduction is not new, and the author written many pages.
Response 1: We understand that the Introduction section may seem “not new” and “too long” regarding the main aspects tackled in the manuscript. However, we think that the potential reader here can be both an adipose-tissue researcher looking for EV research as well as an EV researcher looking for adipose-tissue research. In both cases, technical and biological considerations must be addressed in order to get “the best” of the research. Therefore, we have tried to balance the information provided not only in the introduction section but also in every section to reduce the length to a minimum and refer the readers to more specific reviews when needed.
Point 2: The importance of standardization section is good one.
Point 3: Reproducibility and transparency: the EV-TRACK database also good.
Responses 2 & 3: We thank the Reviewer for the favorable comments.
Point 4: Table 1 is good and I will suggest to author if possible please include one more table about the clinical translational aspect of EVs.
Response 4: We thank the Reviewer for the favorable comments about Table 1. Regarding the clinical translational aspect of EVs, we think that this feature is out of the scope of our review. Moreover, the clinical translational impact of EVs is still on-going in the general EV field and use/target of Ad-EVs is really limited. Nevertheless, we agree that this is an interesting aspect and therefore, we have included some comments in the conclusion section by referring to publications which have underlined the potential translational impact of EVs (lines 713-717 in the revised version of this manuscript). We also refer the Reviewer to the another review published by Rome et al. which is also part of this Special Issue and deepens into the Ad-EV described modulatory functions.
Point 5: Also please include some more figures to highlight the topic more representiative and appealing.
Response 5: We totally agree with the Reviewer and thank for the suggestion. In order to synthetize the scope of our manuscript and make it more appealing, we have included an additional new figure 3. We attach separately here the new figure for the Reviewer´s information.
Point 6: The title is sounds more good than manuscript text, please write comprehensively to streamline the manuscript.
Response 6: We thank for the favorable comments about out title (if good, it will attract the attention of the readers). We have included different comments through the revised version of the manuscript to emphasize the methodological aspects and biological considerations that may influence the results/conclusions of Ad-EV research. Moreover, we think that the section 3 has been considerably improved thanks to the Reviewer´s suggestions about including a new figure. To improve the streamline of the text, a new paragraph has been also included (lines 238-245 in the revised version) as well as additional comments in the conclusions section.

Round 2
Reviewer 3 Report
No more comments